DATA RELEASE

# Genome assembly of the edible jelly fungus *Dacryopinax spathularia (Dacrymycetaceae)*

Hong Kong Biodiversity Genomics Consortium*,†

## ABSTRACT

The edible jelly fungus *Dacryopinax spathularia* (*Dacrymycetaceae*) is wood-decaying and can be commonly found worldwide. It has found application in food additives, given its ability to synthesize long-chain glycolipids, among other uses. In this study, we present the genome assembly of *D. spathularia* using a combination of PacBio HiFi reads and Omni-C data. The genome size is 29.2 Mb. It has high sequence contiguity and completeness, with a scaffold N50 of 1.925 Mb and a 92.0% BUSCO score. A total of 11,510 protein-coding genes and 474.7 kb repeats (accounting for 1.62% of the genome) were predicted. The *D. spathularia* genome assembly generated in this study provides a valuable resource for understanding their ecology, such as their wood-decaying capability, their evolutionary relationships with other fungi, and their unique biology and applications in the food industry.

**Subjects** Genetics and Genomics, Ecology, Evolutionary Biology

**Submitted:** 11 January 2024

* Correspondence on behalf of the consortium: E-mail: jeromehui@cuhk.edu.hk

† Collaborative Authors: Entomological experts who validated the dataset and their affiliations appears at the end of the document

Preprint submitted at https://doi.org/10.1101/2024.01.16.575489

Included in the series: ***Hong Kong Biodiversity Genomics*** (https://doi.org/10.46471/GIGABYTE_SERIES_0006)

## INTRODUCTION

*Dacryopinax spathularia* (*Dacrymycetaceae*, NCBI:txid139277) (Figure 1A) is a brown-rot fungus commonly found on rotting coniferous and broadleaf wood worldwide. This fungus can be easily distinguished by the spathulate shape of its gelatinous fruiting body [1, 2]. Owing to its production of carotenoid pigments as a protection against UV damage, its external appearance is generally orange to yellow [3]. In addition to its ecological role in nutrient recycling, this species is also edible and commonly known as the "sweet osmanthus ear" mushroom in China [4]. Given its ability to synthesise long-chain glycolipids under fermentation, this species has also been cultivated in the food industry to produce natural preservatives for soft drinks [5].

## CONTEXT

Edible jelly fungus *D. spathularia* (*Dacrymycetaceae*), which was first described as *Merulius spathularius*, is a macrofungus basidiomycete and can be commonly found on rotting coniferous and broadleaf wood in tropics and subtropics. Its wood-decaying ability facilitates nutrient recycling in forest ecosystems [6]. This species is edible and frequently cultivated in industry to produce food additives such as natural preservatives for soft drinks [4, 7]. In addition, the isolated fungal extract can also display anti-bacterial properties [8]. *D. spathularia* can be naturally found in Asia, Africa, America, Australia and other parts of the Pacific region. To date, the genomic data of the genus *Dacryopinax* is limited to *Dacryopinax primogenitus*, which is used for studying the origin of genes involved in lignin decomposition among different wood-decaying fungi lineages [9]. However, the genomic data of *D. spathularia* is not available.

In Hong Kong, *D. spathularia* can be commonly found [10] and has been selected as one of the species to be sequenced by the Hong Kong Biodiversity Genomics Consortium (also known as EarthBioGenome Project Hong Kong) formed by investigators from eight publicly funded universities. Here, we present the genome assembly of *D. spathularia*, which was assembled from PacBio long reads and Omni-C sequencing data. The *D. spathularia* genome will help better understand this fungus' ecology, the genetic basis of its wood-decaying ability, the phylogenetic relationships in its family and the biosynthesis of the long-chain glycolipids that are used as natural preservatives in the food industry.

## METHODS

### Sample collection and culture of fungal isolates

The fruiting bodies of *D. spathularia* were collected in Luk Keng, Hong Kong, on 20 June 2022 (Figure 1A). The fungal isolate was transferred from the edge of fruit bodies to potato dextrose agar (BD Difco™) plates using a pair of sterilized forceps. The remaining collected fruit bodies were snap-frozen with liquid nitrogen and stored in a −80 °C freezer. Fungal hyphae from 2-week-old colonies were transferred to new plates for purification for at least three rounds. The identity of the isolate, termed "F14", was assigned by DNA barcoding using the sequence of the Translation elongation factor 1 alpha (TEF-1α) gene using the primer pairs EF1-1018F and EF1-1620R [11] (Figure 1B).

### High molecular weight DNA extraction

Approximately 1.5 g of mycelia of *D. spathularia* isolate was collected from the upper layer of the agar culture and ground in a mortar with liquid nitrogen. High molecular weight (HMW) genomic DNA was isolated with cetyltrimethylammonium bromide (CTAB) treatment, followed by the NucleoBond HMW DNA kit (Macherey Nagel Item No. 740160.20). Briefly, the ground tissue was transferred to 5 mL CTAB buffer [12] with an addition of 1% Polyvinylpyrrolidone for 1 hour digestion at 55 °C. After RNAse A treatment, by adding 100 µL RNAse A and incubating for 10 min at room temperature, 1.6 mL 3M potassium acetate was added to the lysate. The lysate was then aliquoted into six 2 mL tubes (each containing ~1.1 mL lysate). Next, 800 µL chloroform:IAA (24:1) was added to each tube and gently mixed by inverting the tubes for ~10 s, followed by centrifugation at >10,000 × *g* for 5 min. The supernatant (~900 µL from each tube) was transferred to a new tube and 800 µL chloroform:IAA (24:1) was added for another round of wash with the same procedure of mixing and centrifugation. Subsequently, the supernatant (~800 µL from each tube, ~4.8 mL total) was mixed with ~1.2 mL H1 buffer from the NucleoBond HMW DNA kit for a final volume of 6 mL and processed according to the manufacturer's protocol. The DNA sample was eluted in 80 µL elution buffer (PacBio Ref. No. 101-633-500) and its quantity and quality were assessed with NanoDrop™ One/OneC Microvolume UV–Vis Spectrophotometer, Qubit® Fluorometer, and overnight pulse-field gel electrophoresis (Figure 1C).

### PacBio library preparation and sequencing

DNA shearing was first performed from 5 µg HMW DNA in 120 µL elution buffer using a g-tube (Covaris Part No. 520079) with six centrifugation steps at 1,990 × *g* for 2 min. The sheared DNA sample was purified using SMRTbell® cleanup beads (PacBio Ref. No. 102158-300). Also, 2 µL of the DNA sample was used for quality check through overnight pulse-field gel electrophoresis and Qubit® Fluorometer quantification. An SMRTbell library



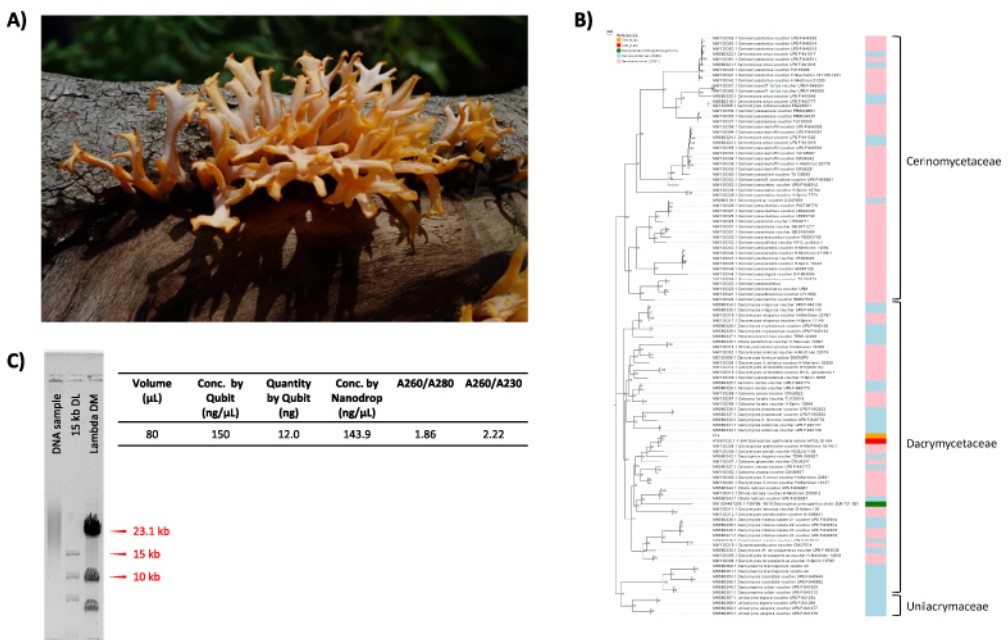

**Figure 1.** Sample information of *D. spathularia*. (A) Picture of the *D. spathularia* we collected in the field; (B) Phylogenetic analysis of the TEF-1α gene region in the *D. spathularia* fungal isolate "F14" of this study. Sequences from related phylogenetic studies and NCBI accessions were incorporated, including Zaroma & Ekman [23], Savchenko *et al.* [24], the *D. primogenitus* genome (NW_024467206.1:736197-736766) and the NCBI BLAST result of a *D. spathularia* isolate (Accession: AY881020.1), which are highlighted in light blue, pink, green, and red, respectively. The fungal isolate "F14" is highlighted in orange. The bootstrap percentage values are shown at the nodes. The sequence alignment and tree file can be retrieved from [29]; (C) Information on the quality control of the extracted high molecular weight DNA sample. The left panel shows the photograph of the overnight pulse-field gel electrophoresis of the extracted DNA sample together with a 15 kb DNA ladder (15 kb DL) and Lambda-Hind III Digest marker (Lambda DM). The right panel summarizes the information on Qubit® Fluorometer and NanoDrop™ One/OneC Microvolume UV–Vis Spectrophotometer. Abbreviation: conc., concentration.

was then prepared by following the protocol of the SMRTbell® prep kit 3.0 (PacBio Ref. No. 102-141-700). Briefly, the sheared DNA was repaired and polished at both ends, followed by A-tailing and ligation of T-overhand SMRTbell adapters. A subsequent purification step was processed with SMRTbell® cleanup beads and 2 μL of sample were used for quality check as mentioned above. Nuclease treatment was then used to remove non-SMRT bell structures. A final size-selection step using 35% AMPure PB beads was processed to eliminate short fragments.

The final library preparation was performed with The Sequel® II binding kit 3.2 (PacBio Ref. No. 102-194-100) before sequencing. The SMRTbell library followed by annealing and binding with Sequel II® primer 3.2 and Sequel II® DNA polymerase 2.2, respectively. SMRTbell® cleanup beads were used to further clean up the library, to which diluted Sequel II® DNA Internal Control Complex was added. The final library was loaded at an on-plate concentration of 90 pM with the diffusion loading mode. Sequencing was performed on the Pacific Biosciences SEQUEL IIe System for a run of 30-hour movies with 120 min pre-extension to output highly accurate long reads (HiFi) reads with one SMRT cell. Details of the resulting sequencing data are listed in Table 1.

**Table 1.** Summary of the genome and transcriptome sequencing information.

| Library | Reads | Bases | Coverage (X) | Accession number |
|---|---|---|---|---|
| PacBio HiFi | 757,219 | 9,340,525,111 | 319 | SRR24631918 |
| Omnic | 23,101,878 | 3,465,281,700 | 118 | SRR27412332 |
| mRNA | 39,483,524 | 5,922,432,049 | 202 | SRR27412333 |

## Omnic-C library preparation and sequencing

Approximately 0.5 g of stored fruit body was ground to powder with liquid nitrogen and used for the construction of an Omni-C library by following the plant tissue protocol for the Dovetail® Omni-C® Library Preparation Kit (Dovetail Cat. No. 21005). The ground tissue was transferred to 4 mL 1× PBS and subjected to crosslinking with formaldehyde and digestion with endonuclease DNase I. The quantity and fragment size of the lysate were assessed with Qubit® Fluorometer and TapeStation D5000 HS ScreenTape, respectively. The qualified lysate was polished at the DNA ends and ligated with biotinylated bridge adaptors, followed by proximity ligation, crosslink reversal of DNA and purification with SPRIselect™ Beads (Beckman Coulter Product No. B23317). The end repair and adapter ligation were performed with the Dovetail™ Library Module for Illumina (Dovetail Cat. No. 21004). The library was then sheared with USER Enzyme Mix and purified with SPRIselect™ Beads. The DNA fragments were isolated in Streptavidin Beads, from which the library was amplified with Universal and Index PCR Primers from the Dovetail™ Primer Set for Illumina (Dovetail Cat. No. 25005). Size selection, targeting fragment sizes between 350 bp and 1000 bp, was performed with SPRIselect™ Beads. The quantity and fragment size of the library were assessed by Qubit® Fluorometer and TapeStation D5000 HS ScreenTape, respectively. The resulting library was sequenced on an Illumina HiSeq-PE150 platform. Details of the resulting sequencing data are listed in Table 1.

## RNA extraction and transcriptome sequencing

Approximately 1 g of mycelia of *D. spathularia* isolate was ground in a mortar with liquid nitrogen. Total RNA was isolated from the ground tissue using the mirVana miRNA Isolation Kit (Ambion), following the manufacturer's instructions. The RNA sample underwent quality control with NanoDrop™ One/OneC Microvolume UV–Vis Spectrophotometer and 1% agarose gel electrophoresis. Finally, the qualified sample was sent to Novogene Co. Ltd (Hong Kong, China) for 150 bp paired-end sequencing. Details of the resulting sequencing data are listed in Table 1.

## Genome assembly and gene model prediction

A *de novo* genome assembly was conducted with Hifiasm [13], which was screened with BlobTools (v1.1.1) [14] by searching against the NT database using blastn (RRID:SCR_004870) to identify and remove any possible contaminations. Haplotypic duplications were discarded using purge_dups (RRID:SCR_021173) according to the depth of the HiFi reads [15]. The Omni-C data were used to scaffold the assembly using YaHS [16].

A gene model prediction was performed using funannotate (RRID:SCR_023039) [17]. RNA sequencing data were first processed using Trimmomatic (v0.39; RRID:SCR_011848) [18] and Kraken2 (v2.0.8 with kraken2 database k2_standard_20210517; RRID:SCR_005484) [19] to remove low quality and contaminated reads. The processed reads were then aligned to the soft-masked repeat genome using HISAT2 (RRID:SCR_015530) to run the

genome-guided Trinity (RRID:SCR_013048) [20] with parameters "--stranded RF --jaccard_clip". This step generated 44,384 transcripts. Gene models were then predicted together with the protein evidence from *Dacryopinax primogenitus* (GCF_000292625.1) [9] using funannotate with parameters "--protein_evidence GCF_000292625.1_Dacryopinax_sp._DJM_731_SSP1_v1.0.proteins.faa --genemark_mode ET -- optimize_augustus --busco_db dikarya --organism fungus -d --max_intronlen 3000". The Trinity transcript alignments were converted to the GFF3 format and input to PASA (RRID:SCR_014656) alignment in the Launch_PASA_pipeline.pl process to generate the PASA models trained by TransDecoder (RRID:SCR_017647), followed by the selection of the PASA gene models using the kallisto (RRID:SCR_016582) TPM data. The PASA gene models were then used for training Augustus (RRID:SCR_008417) in the funannotate-predict step. The gene models from several prediction sources, with a total of 54,275 genes from Augustus (4,967), HiQ (4,624), CodingQuarry (11,762), GlimmerHMM (RRID:SCR_002654) (10,843), PASA (11,217), SNAP (RRID:SCR_007936) (10,862), were passed to Evidence Modeler with EVM Weights "'Augustus': 1, 'HiQ': 2, 'CodingQuarry': 2, 'GlimmerHMM': 1, 'pasa': 6, 'snap': 1, 'proteins': 1, 'transcripts': 1" to generate the gene model annotation files. Untranslated regions (UTRs) were then captured in the funannotate-update step using PASA to generate the final genome annotation files.

### Repeat annotation

Transposable element (TE) annotation was performed by following the Earl Grey TE annotation workflow pipeline (version 1.2) [21].

## RESULTS AND DISCUSSION

### Genome assembly

A total of 9.34 Gb HiFi reads was generated from PacBio sequencing (Table 1). After scaffolding with 3.46 Gb Omni-C data, the *D. spathularia* genome assembly has a size of 29.2 Mb, scaffold N50 of 1.925 Mb and 92.0% BUSCO (RRID:SCR_015008) score [22] (Figure 2 and Table 2), and 19 out of 24 scaffolds are >100 kb in length and validated by inspection of the Omni-C contact maps (Figure 1C and Table 2). The genome size is similar to *D. primogenitus* (29.5 Mb) [9] (Figure 1B) and GenomeScope (RRID:SCR_017014) estimated heterozygosity of 5.09% (Figure 1D; Table 3). Gene model prediction generated a total of 11,510 protein-coding genes with an average protein length of ~451 amino acids and a BUSCO score of 91.9%.

### Repeat content

Repeat content analysis showed that transposable elements (TEs) account for 1.62% of the *D. spathularia* genome (Figure 1E; Tables 4 and 5). The major classified TE was long terminal repeats retransposons (0.95%) and DNA transposons (0.12%) (Tables 4 and 5).

## CONCLUSION AND FUTURE PERSPECTIVE

The study presents the genome assembly of *D. spathularia*, a useful resource for further phylogenomic studies in the family *Dacrymycetaceae* and investigations on the biosynthesis of glycolipids with potential applications in the food industry.

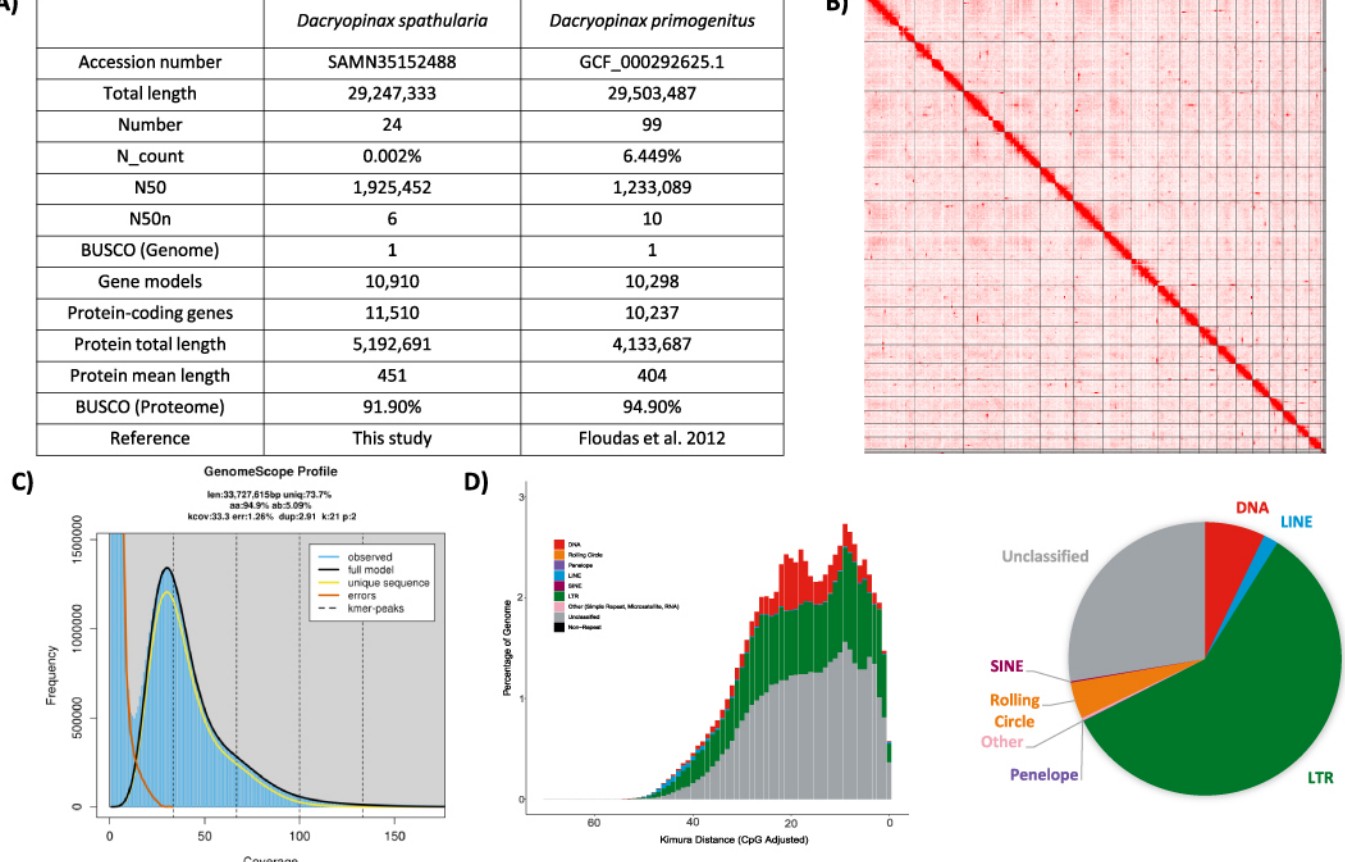

**Figure 2.** (A) Genome statistics; (B) Omni-C contact map of the assembly. The chromatin contact intensities are indicated in red along the matrix of genomic regions, and the boundary of scaffolds are defined by black lines; (C) GenomeScope report summary; (D) Repeat landscape plot (left) and the proportion (right) of repetitive elements in the assembled genome.

## DATA VALIDATION AND QUALITY CONTROL

The identity of the fungal isolate of *D. spathularia* was validated with the DNA barcoding of the TEF-1α gene, which was compared with sequences from phylogenetic studies of *Dacrymycetaceae* [23] and their sister family Cerinomycetaceae [24], the *D. primogenitus* genome (Accession: NW_024467206.1:736197-736766) and *D. spathularia* (Accession: AY881020.1). The sequences were aligned with MAFFT (v7.271; RRID:SCR_011811) [25]. A phylogenetic tree was constructed with FastTree [26] with 1,000 bootstraps and visualized in Evolview v3 [27]. The *D. spathularia* isolate in this study was clustered with another two *D. spathularia* accessions with a bootstrap support of 92/100 (Figure 1B).

For HMW DNA extraction and Pacbio library preparation, the samples were subject to quality control with NanoDrop™ One/OneC Microvolume UV–Vis Spectrophotometer, Qubit® Fluorometer, and overnight pulse-field gel electrophoresis (Figure 1C). The quality of the Omni-C library was inspected with Qubit® Fluorometer and TapeStation D5000 HS ScreenTape.

During the genome assembly, BlobTools (v1.1.1) [14] was employed to identify and remove any possible contaminations (Figure 3). The assembled genome and gene model



**Table 2.** Genome statistics and sequencing information.

| | *Dacryopinax spathularia* | *Dacryopinax sp* |
|---|---|---|
| Accession number | SAMN35152488 | GCF_000292625.1 |
| Total length | 29,247,333 | 29,503,487 |
| Number | 24 | 99 |
| Mean_length | 1,218,639 | 298,015 |
| Longest | 3,153,799 | 2,114,445 |
| Shortest | 21,000 | 2,007 |
| N_count | 0.002% | 6.449% |
| Gaps | 3 | 880 |
| N50 | 1,925,452 | 1,233,089 |
| N50n | 6 | 10 |
| N70 | 1,224,341 | 961,128 |
| N70n | 10 | 15 |
| N90 | 879,151 | 680,945 |
| N90n | 15 | 23 |
| BUSCO (fungi_odb10, Genome) | C:92.0% [S:91.3%, D:0.7%], F:0.9%, M:7.1%, n:758 | C:93.9% [S:93.5%, D:0.4%], F:1.8%, M:4.3%, n:758 |
| Gene models | 10,910 | 10,298 |
| Protein-coding genes | 11,510 | 10,237 |
| Protein total length | 5,192,691 | 4,133,687 |
| Protein mean length | 451 | 404 |
| BUSCO (fungi_odb10, Proteome) | C:91.9% [S:86.8%, D:5.1%], F:1.7%, M:6.4%, n:758 | C:94.9% [S:94.1%, D:0.8%], F:1.3%, M:3.8%, n:758 |

**Table 3.** Information on scaffold names and lengths.

| Scaffold number | Scaffold name | Scaffold length |
|---|---|---|
| 1 | scaffold_1 | 3,153,799 |
| 2 | scaffold_2 | 3,126,553 |
| 3 | scaffold_3 | 2,613,975 |
| 4 | scaffold_4 | 2,217,289 |
| 5 | scaffold_5 | 2,133,437 |
| 6 | scaffold_6 | 1,925,452 |
| 7 | scaffold_7 | 1,733,428 |
| 8 | scaffold_8 | 1,704,084 |
| 9 | scaffold_9 | 1,371,616 |
| 10 | scaffold_10 | 1,224,341 |
| 11 | scaffold_11 | 1,181,679 |
| 12 | scaffold_12 | 1,119,408 |
| 13 | scaffold_13 | 1,078,888 |
| 14 | scaffold_14 | 1,063,181 |
| 15 | scaffold_15 | 879,151 |
| 16 | scaffold_16 | 827,634 |
| 17 | scaffold_17 | 815,264 |
| 18 | scaffold_18 | 776,457 |
| 19 | scaffold_19 | 146,554 |
| 20 | scaffold_21 | 47,886 |
| 21 | scaffold_22 | 36,629 |
| 22 | scaffold_23 | 28,000 |
| 23 | scaffold_24 | 21,628 |
| 24 | scaffold_25 | 21,000 |

prediction were assessed with BUSCO (v5.5.0) [22] using the fungal dataset fungi_odb10. Finally, GenomeScope2 [28] was used to estimate the genome size and heterozygosity of the assembly.

**Table 4.** Summary of the GenomeScope statistics.

| Property | Min | Max |
|---|---|---|
| Homozygous (aa) | 94.88% | 94.94% |
| Heterozygous (ab) | 5.06% | 5.12% |
| Genome haploid length (bp) | 33,574,999 | 33,727,615 |
| Genome repeat length (bp) | 8,842,276 | 8,882,469 |
| Genome unique length (bp) | 24,732,723 | 24,845,146 |
| Model fit | 86.24% | 97.69% |
| Read error rate | 1.26% | 1.26% |

GenomeScope version 2.0. $p = 2$, $k = 21$.

**Table 5.** Summary of the TE annotations.

| Classification | Total length (bp) | Count | Proportion (%) | No. of distinct classifications |
|---|---|---|---|---|
| DNA | 34,030 | 83 | 0.1164 | 70 |
| LINE | 8,291 | 35 | 0.0283 | 32 |
| LTR | 278,072 | 148 | 0.9508 | 76 |
| Other (Simple Repeat, Microsatellite, RNA) | 320 | 2 | 0.0011 | 2 |
| Penelope | 1,559 | 9 | 0.0053 | 9 |
| Rolling Circle | 20,551 | 22 | 0.0703 | 15 |
| SINE | 702 | 5 | 0.0024 | 2 |
| Unclassified | 131,188 | 154 | 0.4485 | 126 |
| **SUM** | **474,713** | **458** | **1.6231** | **332** |

## DATA AVAILABILITY

The raw reads generated in this study were deposited in the NCBI database under the SRA accessions SRR24631918, SRR27412332 and SRR27412333. The GenomeScope report, genome, genome annotation and repeat annotation files were made publicly available in Figshare [29].

## ABBREVIATIONS

CTAB, cetyltrimethylammonium bromide; HiFi, highly accurate long reads; HMW, high molecular weight; TE, transposable element; TEF-1α, Translation elongation factor 1 alpha.

## DECLARATIONS

### Ethics approval and consent to participate

The authors declare that ethical approval was not required for this type of research.

### Competing interests

The authors declare that they do not have competing interests.

### Authors' contribution

JHLH, TFC, LLC, SGC, CCC, JKHF, JDG, SCKL, YHS, CKCW, KYLY and YW conceived and supervised the study. TKC and STSL collected the samples and carried out DNA extraction, library preparation and genome sequencing. HYY arranged the logistics of samples. WN performed genome assembly and gene model prediction.

**A)**

PacBio.Primary.purged.fa.blobtools_pacbio.blobDB.json.bestsum.phylum.p8.span.100.blobplot.bam0

Basidiomycota (24:29.2MB;1,925,452nt)
Pseudomonadota (1:0.07MB;65,528nt)
Ascomycota (1:0.05MB;50,408nt)

**B)**

**Figure 3.** Genome assembly quality control and contaminant/cobiont detection. (A) BlobPlot of the assembly. Each circle represents a scaffold with its size proportional to the scaffold length. The colour of the circle indicates the taxonomic assignment based on the BLAST similarity search results; (B) ReadCovPlot of the assembly showing the proportion of unmapped and mapped sequences (left panel). The latter is further described according to the rank of the phylum (right panel).

## Funding

This work was funded and supported by the Hong Kong Research Grant Council Collaborative Research Fund (C4015-20EF), CUHK Strategic Seed Funding for Collaborative Research Scheme (3133356) and CUHK Group Research Scheme (3110154).

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

## DETAILS OF COLLABORATIVE AUTHORS

### • List of authors in Hong Kong Biodiversity Genomics Consortium

Jerome H. L. Hui,[1] Ting Fung Chan,[2] Leo Lai Chan,[3] Siu Gin Cheung,[4] Chi Chiu Cheang,[5,6] James Kar-Hei Fang,[7] Juan Diego Gaitan-Espitia,[8] Stanley Chun Kwan Lau,[9] Yik Hei Sung,[10,11] Chris Kong Chu Wong,[12] Kevin Yuk-Lap Yip,[13,14] Yingying Wei,[15] Tze Kiu Chong,[1] Sean Tsz Sum Law,[1] Wenyan Nong,[1] Ho Yin Yip[1]

[1]School of Life Sciences, Simon F.S. Li Marine Science Laboratory, State Key Laboratory of Agrobiotechnology, Institute of Environment, Energy and Sustainability, The Chinese University of Hong Kong, Hong Kong, China

[2]School of Life Sciences, State Key Laboratory of Agrobiotechnology, The Chinese University of Hong Kong, Hong Kong SAR, China

[3]State Key Laboratory of Marine Pollution and Department of Biomedical Sciences, City University of Hong Kong, Hong Kong SAR, China

[4]State Key Laboratory of Marine Pollution and Department of Chemistry, City University of Hong Kong, Hong Kong SAR, China

[5]Department of Science and Environmental Studies, The Education University of Hong Kong, Hong Kong SAR, China

[6]EcoEdu PEI, Charlottetown, PE, C1A 4B7, Canada

[7]Department of Food Science and Nutrition, Research Institute for Future Food, and State Key Laboratory of Marine Pollution, The Hong Kong Polytechnic University, Hong Kong SAR, China

[8]The Swire Institute of Marine Science and School of Biological Sciences, The University of Hong Kong, Hong Kong SAR, China

[9]Department of Ocean Science, The Hong Kong University of Science and Technology, Hong Kong SAR, China

[10]Science Unit, Lingnan University, Hong Kong SAR, China

[11]School of Allied Health Sciences, University of Suffolk, Ipswich, IP4 1QJ, UK

[12]Croucher Institute for Environmental Sciences, and Department of Biology, Hong Kong Baptist University, Hong Kong SAR, China

[13]Department of Computer Science and Engineering, The Chinese University of Hong Kong, Hong Kong SAR, China

[14]Sanford Burnham Prebys Medical Discovery Institute, La Jolla, CA, USA

[15]Department of Statistics, The Chinese University of Hong Kong, Hong Kong SAR, China

