## [Editor Report]

Editor’s AssessmentThis work is part of a series of papers from the Hong Kong Biodiversity Genomics Consortium sequencing the rich biodiversity of species in Hong Kong. This example This work is part of a series of papers from the Hong Kong Biodiversity Genomics Consortium sequencing the rich biodiversity of species in Hong Kong. This example presenting the first whole genome assembly of Dacryopinax spathularia, an edible mushroom-forming fungus that is used in the food industry to produce natural preservatives. Using PacBio and Omni-C data a 29.2 Mb genome was assembled, with a scaffold N50 of 1.925 Mb and 92.0% BUSCO score demonstrating the quality (review pushing the authors to provide more detail and QC stats to help better convince on this). This data providing a useful resource for further phylogenomic studies in the family Dacrymycetaceae and investigations on the biosynthesis of glycolipids with potential applications in the food industry.

---

## [Reviewer Report]

Reviewer name and names of any other individual's who aided in reviewer Anton SonnenbergDo you understand and agree to our policy of having open and named reviews, and having your review included with the published papers. (If no, please inform the editor that you cannot review this manuscript.)YesIs the language of sufficient quality?YesPlease add additional comments on language quality to clarify if needed
-Are all data available and do they match the descriptions in the paper? YesAdditional CommentsAre the data and metadata consistent with relevant minimum information or reporting standards? See GigaDB checklists for examples <a href="http://gigadb.org/site/guide" target="_blank">http://gigadb.org/site/guide</a>YesAdditional CommentsIs the data acquisition clear, complete and methodologically sound?YesAdditional CommentsIs there sufficient detail in the methods and data-processing steps to allow reproduction?YesAdditional CommentsIs there sufficient data validation and statistical analyses of data quality? YesAdditional CommentsIs the validation suitable for this type of data?YesAdditional CommentsIs there sufficient information for others to reuse this dataset or integrate it with other data?YesAdditional CommentsFigure 1E could be improved by eliminating in the pie-chart the non-repeat sequences or bar-plot the repeats. That will visualize better the frequencies of each type of repeats.Any Additional Overall Comments to the AuthorRecommendationMinor Revision

---

## [Reviewer Report]

Upload additional filesDRR-202401-04/form/20240216_review-comments-RIA.docxReviewer name and names of any other individual's who aided in reviewer Riccardo IacovelliDo you understand and agree to our policy of having open and named reviews, and having your review included with the published papers. (If no, please inform the editor that you cannot review this manuscript.)YesIs the language of sufficient quality?NoPlease add additional comments on language quality to clarify if needed
There are several typos spread across the text, and some sentences are written in an unclear manner. I provide some suggestions in the attachment.Are all data available and do they match the descriptions in the paper? YesAdditional CommentsYes, but some of the data shown is rather unclear and/or not supported by sufficient explanation. For example, what is actually Fig. 1C showing? Because the reference in the text (which contains a typo, line 197) refers to something else.  What is the second set of stats in Fig. 1B? This other organism is not mentioned at all anywhere in the manuscript.Are the data and metadata consistent with relevant minimum information or reporting standards? See GigaDB checklists for examples <a href="http://gigadb.org/site/guide" target="_blank">http://gigadb.org/site/guide</a>NoAdditional CommentsNCBI TaxID of the sequenced species object of this work is missing.Is the data acquisition clear, complete and methodologically sound?YesAdditional CommentsSee below.Is there sufficient detail in the methods and data-processing steps to allow reproduction?NoAdditional CommentsIn my opinion, some of the procedures described for the processing of the sample and library prep for sequencing are reported in an unclear way. For example, lines 100-103: no details on RNAse A treatment; how do you define chloroform:IAA (24:1) washes? how much supernatant is added to how much H1 buffer to have the final volume of 6 ml? Another example, lines 180-175: what parameters did you use for EvidenceModeler to generate the final consensus genes model? The weight given to each particular prediction set is important. Is there sufficient data validation and statistical analyses of data quality? NoAdditional CommentsWhile sufficient data validation and statistical analyses have been carried out with respect to DNA sequencing and genome assembly, nothing is reported about DNA extraction and quality. The authors mention several times throughout the text that DNA preps are checked via NanoDrop, Qubit, gel electrophoresis, etc. But none of this is shown in the main body or in the supplementary information. Without this information, it is difficult to assess directly the efficacy of DNA extraction and preparation methods. I recommend including this type of data.Is the validation suitable for this type of data?YesAdditional CommentsAside from comments above, yes.Is there sufficient information for others to reuse this dataset or integrate it with other data?YesAdditional CommentsAny Additional Overall Comments to the AuthorSee attached document for other comments.RecommendationMinor Revision